# Circulating Malondialdehyde Is a Potential Biomarker for Predicting All-Cause Mortality during Follow-Up by Reflecting Comprehensive Inflammation at Diagnosis in Patients with Antineutrophil Cytoplasmic Antibody-Associated Vasculitis

**DOI:** 10.3390/medicina60071182

**Published:** 2024-07-21

**Authors:** Jihye Chung, Taejun Yoon, Hyunsue Do, Yong-Beom Park, Sang-Won Lee

**Affiliations:** 1Division of Rheumatology, Department of Internal Medicine, Yonsei University College of Medicine, Seoul 03722, Republic of Korea; jchung0831@yuhs.ac (J.C.); yongbpark@yuhs.ac (Y.-B.P.); 2Department of Medical Science, BK21 Plus Project, Yonsei University College of Medicine, Seoul 03722, Republic of Korea; tjyoonn92@gmail.com; 3Division of Rheumatology, Department of Internal Medicine, Kangwon National University School of Medicine, Chuncheon 24289, Republic of Korea; dohs712@yuhs.ac; 4Institute for Immunology and Immunological Diseases, Yonsei University College of Medicine, Seoul 03722, Republic of Korea

**Keywords:** malondialdehyde, antineutrophil cytoplasmic antibody, vasculitis, inflammation, mortality

## Abstract

*Background and Objectives:* To investigate whether circulating malondialdehyde (cMDA) at diagnosis could contribute to reflecting cross-sectional comprehensive inflammation or vasculitis activity and further predicting all-cause mortality during follow-up in patients with antineutrophil cytoplasmic antibody-associated vasculitis (AAV). *Materials and Methods:* This study included 78 patients with AAV. Erythrocyte sedimentation rate (ESR) and C-reactive protein (CRP) levels were collected as indices reflecting cross-sectional comprehensive inflammation, whereas the Birmingham vasculitis activity score (bVAS), and the five-factor score (FFS) were reviewed as AAV-specific indices. All-cause mortality was considered to be a poor outcome during follow-up. cMDA was measured from stored sera. *Results:* The median age of the 78 patients (32 men and 46 women) was 63.0 years. The median BVAS, FFS, ESR, and CRP were 5.0, 0, 24.5 mm/h, and 3.4 mg/L, respectively. Six patients died during the median follow-up duration based on all-cause mortality at 26.7 months. At diagnosis, cMDA was significantly correlated with cross-sectional ESR but not with BVAS or FFS. Compared to patients with cMDA < 221.7 ng/mL, those with cMDA ≥ 221.7 ng/mL at diagnosis exhibited an increased relative risk (RR 12.4) for all-cause mortality and further showed a decreased cumulative patient survival rate. Cox analyses revealed that cMDA ≥ 221.7 ng/mL (hazard ratio 24.076, *p* = 0.007) exhibited an independent association with all-cause mortality during follow-up in patients with AAV. *Conclusions:* cMDA at diagnosis may be a potential biomarker for predicting all-cause mortality during follow-up by reflecting comprehensive inflammation at diagnosis in patients with AAV.

## 1. Introduction

Malondialdehyde is a compound derived from the adducts of peroxidation of polyunsaturated fatty acids and is mainly metabolised into carbon dioxide and water by enzymes such as mitochondrial aldehyde dehydrogenase [1]. Malondialdehyde also forms compounds through covalent bonds with nucleic acids or proteins, which can act as new autoantigens, leading to the initiation and acceleration of various inflammatory signal transductions via the nuclear factor kappa-light-chain enhancer of activated B cells (NF-kB), p38 mitogen-activated protein kinases (MAPK), and extracellular signal-regulated kinase (ERK) pathways [1]. These inflammatory signals may disrupt immune tolerance and cause the development and progression of autoimmune-based inflammatory diseases, in which oxidative stress and the production of reactive oxygen species (ROS) are augmented, resulting in a vicious cycle that increases malondialdehyde production [1,2]. Based on these chain reactions, to date, there have been many efforts in real-world clinical practice to determine the clinical role of circulating malondialdehyde (cMDA) as a biomarker in diverse chronic inflammatory or cancerous diseases [1,3].

Antineutrophil cytoplasmic antibody (ANCA)-associated vasculitis (AAV) is a group of small vessel vasculitis characterised by fibrinoid necrotic inflammation in small vessels without definitive immune deposits on biopsy. According to clinical features at diagnosis, AAV is categorised into three subtypes: microscopic polyangiitis (MPA); granulomatosis with polyangiitis (GPA); and eosinophilic GPA (EGPA) [4,5]. Among the various immune mechanisms involved in the pathogenesis of AAV, similar to malondialdehyde, inflammatory signalling pathways, oxidative stress, and ROS production are known to play important roles [6,7,8]. Theoretically, therefore, cMDA may be a potential biomarker for estimating the scale of oxidative stress and the degree of comprehensive inflammation in patients with AAV. To date, however, no study has clarified the clinical implications of cMDA in patients with AAV. As such, in this study, we investigated whether cMDA at diagnosis contributes to reflecting cross-sectional comprehensive inflammation or vasculitis activity and further predicting all-cause mortality during follow-up in patients with AAV.

## 2. Materials and Methods

### 2.1. Patients

We randomly selected 80 patients with AAV from the Severance Hospital ANCA-associated VasculitidEs (SHAVE) cohort, an observational cohort of Korean patients with AAV according to the inclusion criteria as follows: First of all, patients were diagnosed with AAV at the Division of Rheumatology, Department of Internal Medicine, Yonsei University College of Medicine, and Severance Hospital for the first time. They met the 2007 European Medicines Agency algorithms for AAV, the revised 2012 International Chapel Hill Consensus Conference Nomenclature of Vasculitides, and the 2022 American College of Rheumatology/European Alliance of Associations for Rheumatology (ACR/EULAR) classification criteria for MPA, GPA, and GPA [4,5,9,10,11,12]. Their medical records were well-documented enough to collect detailed clinical, laboratory, radiological, and histological data not only at diagnosis but also during follow-up to the last visit. Particularly, laboratory and radiological test results were recognised only when their tests were performed within less than 2 weeks at the time of AAV diagnosis. They were followed up for at least six months after AAV diagnosis. They had no serious medical conditions mimicking AAV such as malignancies or severe infectious diseases requiring hospitalisation as described in the 2022 ACR/EULAR criteria at the time of AAV diagnosis [9,10,11]. They were not exposed to immunosuppressive drugs for AAV treatment within at least four weeks (one month) before AAV diagnosis; however, glucocorticoids in less than 10 mg of dose equivalent to prednisolone were allowed for relieving AAV symptoms [13,14]. Of the 80 patients screened, 2 were excluded due to errors in measuring cMDA; as such, data from 78 patients were ultimately included and analysed.

The present study was approved by the Institutional Review Board (IRB) of Severance Hospital, Seoul, Republic of Korea (IRB number 4-2016-0901), and conducted in accordance with the Declaration of Helsinki. All patients in this study provided written informed consent upon enrolment in the SHAVE cohort (at the time of AAV diagnosis and blood sampling). The IRB waived the requirements for additional written informed consent when it was obtained upon enrolment into the SHAVE cohort.

### 2.2. Clinical Data

Regarding data at diagnosis, age, sex, smoking history, and body mass index were collected as demographic ones. In terms of AAV-specific variables, subtype, ANCA type, the Birmingham vasculitis activity score (BVAS), and the five-factor score (FFS) were reviewed [15,16]. Type 2 diabetes mellitus and hypertension were assessed as comorbidities associated with mortality. Additionally, initial laboratory results, including erythrocyte sedimentation rate (ESR) and C-reactive protein (CRP), were obtained. In addition to myeloperoxidase (MPO)-ANCA and proteinase 3 (PR3)-ANCA, perinuclear (P)-ANCA and cytoplasmic (C)-ANCA were also accepted as ANCA results as recommended by the current criteria [9,10,11,12]. Regarding data during follow-up, the follow-up duration based on all-cause mortality was defined as the period from AAV diagnosis to death for deceased patients and the period from AAV diagnosis to the last visit for surviving patients, respectively. For deceased patients, death means the same as the last visit; therefore, in this study, the follow-up duration in all 78 patients was the same as the follow-up duration based on all-cause mortality.

The patients’ history of administration of glucocorticoids and immunosuppressive drugs was also investigated.

### 2.3. Measurement of cMDA 

Whole-blood samples were collected from all the patients at the time of diagnosis of AAV when written informed consent was obtained. On the same day, sera were immediately isolated from whole blood and stored at −80 °C. cMDA was measured from the stored sera using enzyme-linked immunosorbent assay kits (LSBio, Shirley, MA, USA).

### 2.4. Statistical Analyses 

All statistical analyses were performed using SPSS version 26 (IBM Corporation, Armonk, NY, USA) for Windows (Microsoft Corporation, Redmond, WA, USA). Continuous and categorical variables were expressed as medians (interquartile rage [IQR], i.e., P25-P75 percentiles) and number (percentage). Correlation coefficients (r) between the two variables were determined using Pearson correlation analysis. A significant area under the curve (AUC) was determined using a receiver operating characteristic (ROC) curve analysis. The cut-off was extrapolated by performing the ROC curve analysis and selected as the value with the maximum sum of sensitivity and specificity, while a relative risk (RR) of the cut-off for all-cause mortality was analysed using contingency tables and the chi-square test. A comparison of the cumulative survival rates between the two groups was performed using a Kaplan–Meier survival analysis with the log-rank test. A multivariable Cox proportional hazard model using variables with *p* < 0.1 in a univariable Cox analysis was performed to obtain a hazard ratio (HR) during follow-up. *p* < 0.05 was considered to be statistically significant.

## 3. Results

### 3.1. Patient Characteristics 

Regarding the data at diagnosis, the median age of the 78 patients (32 men and 46 women) was 63.0 years. There were three ex-smokers (3.8%), and the median BMI was 22.4 kg/m^2^. Among the 78 patients, 38, 23, and 17 were classified as having MPA, GPA, and EGPA, respectively. MPO-ANCA (or P-ANCA) and PR3-ANCA (or C-ANCA) were detected in 43 and 12 patients, respectively. A total of 17 and 25 patients had type 2 diabetes mellitus and hypertension, respectively. The median BVAS, FFS, ESR, and CRP were 5.0, 0, 24.5 mm/h, and 3.4 mg/L, respectively. For the laboratory results, the median values were as follows: white blood cell count, 7710.0/mm^3^; haemoglobin, 12.5 g/dL; platelet count, 241,000/mm^3^; fasting glucose, 94.5 mg/dL; total cholesterol, 175.5 mg/dL; blood urea nitrogen, 19.3 mg/dL; serum creatinine, 0.8 mg/dL; total serum protein, 6.8 g/dL; and serum albumin, 4.2 g/dL. Regarding the data during AAV follow-up, six patients died during the median follow-up duration based on all-cause mortality at 26.7 months. Of the 78 patients, 77 (98.7%) received glucocorticoids, and the most frequently administered immunosuppressive drug was cyclophosphamide (65.4%), followed by azathioprine (61.5%) and mycophenolate mofetil (25.6%) (Table 1). 

### 3.2. Correlation Analysis 

At diagnosis, cMDA was significantly correlated with only ESR (r = 0.251, *p* = 0.027). cMDA also tended to be correlated with CRP (r = 0.222) but there was no statistical significance. Conversely, no significant correlations of cMDA with age, MPO-ANCA and PR3-ANCA titres, BVAS, and FFS were observed (Table 2). Additionally, there were no significant differences in cMDA according to the presence of MPO-ANCA (or P-ANCA) (*p* = 0.579) or PR3-ANCA (or C-ANCA) (*p* = 0.531).

### 3.3. Cut-Off, RR, and Survival Rates of cMDA for Mortality

Using the ROC curve analysis, the cut-off of cMDA for all-cause mortality was determined to be 221.7 ng/mL (sensitivity and specificity were 66.7% and 86.1%, respectively) (AUC 0.764, 95% confidence interval [CI] 0.547, 0.981) (Figure 1A). All-cause mortality occurred more frequently in patients with cMDA ≥ 221.7 ng/mL than in those with cMDA < 221.7 ng/mL (28.6% vs. 3.1%, *p* = 0.008), and RR was calculated to be 12.400 (95% CI 2.001, 76.842) (Figure 1B). Additionally, patients with cMDA ≥ 221.7 ng/mL exhibited a significantly reduced cumulative patient survival rate compared to those with cMDA < 221.7 ng/mL (*p* < 0.001) (Figure 1C).

### 3.4. Cox Proportional Analyses for All-Cause Mortality during Follow-Up

As described in the Methods section, variables with *p* < 0.1 were defined as significant in the univariable Cox analysis, in which age (HR 1.098), BVAS (HR 1.077), type 2 diabetes mellitus (HR 3.962), ESR (HR 1.022), CRP (HR 1.019), cMDA (HR 1.006), and cMDA ≥ 221.7 ng/mL (HR 11.098) were associated with all-cause mortality during follow-up. In the multivariable analysis including cMDA, only age (HR 1.169, 95% CI 1.024, 1.334) was independently associated with all-cause mortality during follow-up. cMDA tended to be independently associated with all-cause mortality during follow-up, but this association was not statistically significant. However, the multivariable analysis including cMDA ≥ 221.7 ng/mL revealed that both age (HR 1.132, 95% CI 1.012, 1.266) and cMDA ≥ 221.7 ng/mL (HR 24.076, 95% CI 2.422, 239.368) were independently associated with all-cause mortality during follow-up (Table 3).

## 4. Discussion

The present study investigated whether cMDA at diagnosis could contribute to reflecting cross-sectional comprehensive inflammation or vasculitis activity and further predicting all-cause mortality during follow-up in patients with AAV. Several interesting results were obtained. First, at diagnosis, cMDA exhibited a significant correlation with ESR and a tendency to correlate with CRP and white blood cell count, which are related to comprehensive inflammation. Conversely, no correlation was observed between cMDA and BVAS or FFS. Second, during AAV follow-up, cMDA demonstrated the potential to predict all-cause mortality. cMDA ≥ 221.7 ng/mL exhibited a higher RR for all-cause mortality and a reduced survival rate than cMDA < 221.7 ng/mL in patients with AAV. Third, among six variables that were significant in univariable Cox analysis, in multivariable analysis, only cMDA ≥ 221.7 ng/mL exhibited an independent association with all-cause mortality during follow-up in patients with AAV. Based on these results, it was concluded that circulating malondialdehyde at diagnosis may have the potential to act as a biomarker for predicting all-cause mortality during follow-up but not as that for reflecting cross-sectional BVAS or FFS collected at diagnosis in patients with AAV.

In this study, we determined that patients with cMDA ≥ 221.7 ng/mL exhibited a higher risk for all-cause mortality and a lower survival rate than those with lower levels. We attempted to infer the mechanisms by which cMDA at the time of diagnosis could be associated with all-cause mortality occurring during the disease course using three types of risk factors for all-cause mortality. First, in terms of traditional risk factors for all-cause mortality [17], cMDA was not correlated with age or body mass index, and there were no significant differences according to sex (*p* = 0.113), type 2 diabetes mellitus (*p* = 0.473), or hypertension (*p* = 0.363). Additionally, in terms of AAV-specific risk factors for all-cause mortality [18,19], cMDA exhibited no direct correlational relationship with BVAS, FFS, and MPO-ANCA or PR3-ANCA titres at the same time point. Conversely, in terms of inflammation-related risk factors for all-cause mortality [20], cMDA exhibited a correlation with ESR and a tendency to be correlated with CRP. Therefore, despite a weak correlation between cMDA and ESR and only a tendency of correlation between cMDA and CRP, it could be reasonably speculated that cMDA at diagnosis could act as a biomarker for predicting all-cause mortality by reflecting the extent of comprehensive (non-specific) effects of the inflammation-related risk factors of all-cause mortality.

In the Cox analyses performed, we revealed the potential of cMDA as an independent risk factor for all-cause mortality comparable to age in patients with AAV. Of note, among the six variables, cMDA exhibited a superior association with all-cause mortality compared with ESR and CRP, which are variables that reflect the degree of cross-sectional comprehensive inflammation and are the theoretical background for predicting mortality [21,22]. To clarify this issue, we included only variables with *p* < 0.05, such as ESR, CRP, and cMDA in univariable Cox analysis. In multivariable, cMDA tended to be independently associated with all-cause mortality, but it was not significant (HR 1.005, 95% CI 1.000, 1.011). Conversely, cMDA ≥ 221.7 ng/mL exhibited a significantly independent association with all-cause mortality (HR 13.462, 95% CI 2.235, 81.067) (Table 4). Therefore, we tentatively conclude that cMDA may be better or at least comparable to ESR, CRP, and BVAS in predicting all-cause mortality during follow-up in patients with AAV. 

We made several inferences regarding our hypothesis. First, in terms of cMDA, systemic inflammation derived from AAV may enhance oxidative stress, resulting in increased ROS production and accelerated lipid peroxidation, contributing to an increase in cMDA. cMDA may activate several inflammatory signalling pathways, and in turn, augment the inflammatory burden of AAV, leading to the formation of a vicious circle of enhanced malondialdehyde production [1,2,3]. Conversely, in terms of ESR, ESR may reflect the degree of comprehensive inflammation in patients with AAV as usual; however, it cannot participate in or amplify the vicious circle consisting of cMDA, oxidative stress, and inflammatory signals at all (Appendix A). This inference may explain the discrepancy between cMDA and ESR in their potential to predict all-cause mortality in patients with AAV.

On the other hand, to validate our inferences, first, we compared the variables considered to be related to all-cause mortality between surviving and deceased patients. Deceased patients were older than surviving ones, as was naturally expected (74.5 vs. 62.5 years, *p* = 0.028). ESR (99.0 mm/h vs. 23.5 mm/h, *p* = 0.029) and CRP (51.6 mg/L vs. 3.0 mg/L, *p* = 0.035) levels were significantly elevated in deceased patients compared to surviving patients. Additionally, cMDA was also remarkably higher in deceased patients than in surviving patients (242.1 ng/mL vs. 92.9 ng/mL, *p* = 0.032) (Appendix A). Second, when we compared the variables between patients with cMDA ≥ 221.7 ng/mL and those with cMDA < 221.7 ng/mL, all-cause mortality was found more frequently in patients with cMDA ≥ 221.7 ng/mL than those without (28.6% vs. 3.1%, *p* = 0.008); however, no statistically significant differences in ESR and CRP levels were observed between the two groups (Appendix A).To put it simply, these results showed that in the death–survival comparative analysis, both ESR and cMDA significantly contributed to all-cause mortality; however, in the cMDA cut-off-based comparative analysis, only death showed a significant difference, which is thought to support the inference that ESR might not participate in the cMDA vicious circle with a potential to make a significant contribution to an increase in the proportion of all-cause mortality. Although there was a significant difference in CRP levels between surviving and deceased patients, CRP and cMDA did not show a significant correlation; so, they are not mentioned in the inference-supporting description section here.

The advantage of this study is that it is the first to demonstrate that cMDA at diagnosis may contribute to the prediction of all-cause mortality during follow-up by reflecting cross-sectional comprehensive inflammation in patients with AAV. The present study has several limitations, the first of which is its retrospective design. More specifically, although we used clinical data from patients with AAV enrolled in a prospective and observational cohort, interpretation and analyses were performed retrospectively. Owing to this limitation of the study design of a retrospective study, mortality-related comorbidities other than T2DM and hypertension at the time of AAV diagnosis or before it could not be included in this study. The number of patients was not sufficient to generalise our results or apply them immediately to patients newly diagnosed with AAV in clinical practice. Although there was a need for a subgroup analysis among MPA, GPA, and EGPA patients, due to the small number of patients in this pilot study, the subgroup analysis could not be performed. Furthermore, no direct evidence supporting the hypothesis of altered inflammatory signalling was evident. Lastly, serial results of cMDA during the disease course, particularly, at the time close to death, were not available in this study. However, as a pilot study, this investigation may have clinical significance because it explored possibilities for future studies. We believe that a future prospective study including more patients and concretely assessing inflammatory signals will provide more reliable and dynamic information regarding the clinical implications of cMDA at diagnosis in patients newly diagnosed with AAV.

## 5. Conclusions

We demonstrated that cMDA at diagnosis may be a potential biomarker for predicting all-cause mortality during follow-up by reflecting cross-sectional comprehensive inflammation in patients with AAV for the first time.

## Figures and Tables

**Figure 1 medicina-60-01182-f001:**
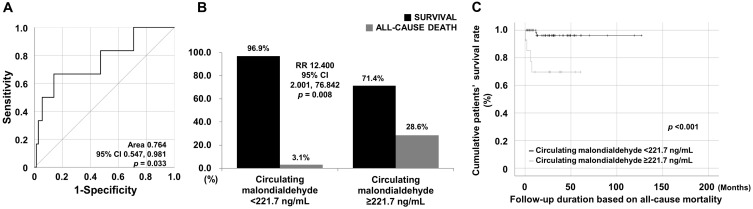
Cut-off, RR, and survival rate of cMDA for mortality. (**A**) Cut-off of cMDA was set as 221.7 ng/mL. (**B**,**C**) cMDA ≥ 221.7 ng/mL exhibited a higher RR for all-cause mortality and a reduced survival rate than cMDA < 221.7 ng/mL in patients with AAV. RR: relative risk; cMDA: circulating malondialdehyde; AAV: ANCA-associated vasculitis; ANCA: antineutrophil cytoplasmic antibody.

**Table 1 medicina-60-01182-t001:** Characteristics of patients with AAV (*n* = 78).

Variables	Values
** *At AAV diagnosis* **	
**Demographic data**	
Age (years)	63.0 (51.8–73.3)
Male sex (*n*, (%))	32 (41.0)
Female sex (*n*, (%))	46 (59.0)
Ex-smoker (*n*, (%))	3 (3.8)
Body mass index (kg/m^2^)	22.4 (20.8–24.7)
**AAV subtypes (*n*, (%))**	
MPA	38 (48.7)
GPA	23 (29.5)
EGPA	17 (21.8)
**ANCA positivity (*n*, (%))**	
MPO-ANCA titre	0 (0–20.5)
PR3-ANCA titre	0 (0–0)
MPO-ANCA (or P-ANCA)-positive	43 (55.1)
PR3-ANCA (or C-ANCA)-positive	12 (15.4)
Both ANCA-positive	3 (3.8)
**AAV-specific indices**	
BVAS	5.0 (3.0–17.0)
FFS	0 (0–1.0)
**Comorbidities (*n*, (%))**	
Type 2 diabetes mellitus	17 (21.8)
Hypertension	25 (32.1)
**Acute-phase reactants**	
ESR (mm/h)	24.5 (9.8–79.8)
CRP (mg/L)	3.4 (0.8–19.4)
**Laboratory results**	
White blood cell count (/mm^3^)	7,710.0 (5952.5–10,525.0)
Haemoglobin (g/dL)	12.5 (10.3–13.6)
Platelet count (x1000/mm^3^)	241.0 (191.5–356.5)
Fasting glucose (mg/dL)	94.5 (87.8–109.3)
Total cholesterol (mg/dL)	175.5 (140.0–212.3)
Blood urea nitrogen (mg/dL)	19.3 (13.7–28.7)
Serum creatinine (mg/dL)	0.8 (0.6–1.6)
Total serum protein (g/dL)	6.8 (6.3–7.3)
Serum albumin (g/dL)	4.2 (3.6–4.4)
**cMDA (ng/mL)**	99.3 (4.0–196.8)
** *During AAV follow-up* **	
**Mortality**	
All-cause mortality	6 (7.7)
Follow-up duration based on all-cause mortality	26.7 (12.0–45.9)
**Medications**	
Glucocorticoids	77 (98.7)
Cyclophosphamide	51 (65.4)
Rituximab	16 (20.5)
Mycophenolate mofetil	20 (25.6)
Azathioprine	48 (61.5)
Tacrolimus	7 (9.0)
Methotrexate	3 (3.8)

Values are expressed as a median (25~75 percentile) or *n* (%). AAV: ANCA-associated vasculitis; ANCA: antineutrophil cytoplasmic antibody; MPA: microscopic polyangiitis; GPA: granulomatosis with polyangiitis; MPO: myeloperoxidase; P: perinuclear; PR3: proteinase 3; C: cytoplasmic; BVAS: the Birmingham vasculitis activity score; FFS: the five-factor score; ESR: erythrocyte sedimentation rate; CRP: C-reactive protein; cMDA: circulating malondialdehyde.

**Table 2 medicina-60-01182-t002:** Correlation analysis of cMDA with continuous variables of demographic data, AAV-specific indices, and acute-phase reactants at diagnosis in patients with AAV.

Variables	cMDA	
	Correlation Coefficient (r)	*p* Value
Age	0.049	0.668
Body mass index	−0.187	0.101
MPO-ANCA titre	0.078	0.498
PR3-ANCA titre	0.006	0.959
BVAS	0.117	0.310
FFS	0.163	0.154
ESR	0.251	0.027
CRP	0.222	0.058

cMDA: circulating malondialdehyde; AAV: ANCA-associated vasculitis; ANCA: antineutrophil cytoplasmic antibody; MPO: myeloperoxidase; P: perinuclear; PR3: proteinase 3; C: cytoplasmic; BVAS: the Birmingham vasculitis activity score; FFS: the five-factor score; ESR: erythrocyte sedimentation rate; CRP: C-reactive protein.

**Table 3 medicina-60-01182-t003:** Cox proportional hazard model analyses of variables at diagnosis for all-cause mortality during follow-up in patients with AAV.

Variables	Univariable			Multivariable (*with cMDA*)			Multivariable (*with cMDA ≥ 221.7* ng/mL)		
	HR	95% CI	*p* Value	HR	95% CI	*p* Value	HR	95% CI	*p* Value
Age	1.098	0.999, 1.207	0.052	1.169	1.024, 1.334	0.021	1.132	1.012, 1.266	0.030
Male sex	2.861	0.524, 15.618	0.225						
Ex-smoker	0.046	0.000, 1,332,560.558	0.726						
Body mass index	1.099	0.862, 1.400	0.446						
MPO-ANCA (or P-ANCA)-positive	4.708	0.549, 40.404	0.158						
PR3-ANCA (or C-ANCA)-positive	0.038	0.000, 516.628	0.501						
BVAS	1.077	0.995, 1.165	0.065	1.028	0.915, 1.155	0.643	1.047	0.910, 1.206	0.520
FFS	2.512	0.831, 5.574	0.115						
Type 2 diabetes mellitus	3.962	0.799, 19.640	0.092	1.650	0.202, 13.460	0.640	2.372	0.226, 24.923	0.472
Hypertension	1.124	0.206, 6.141	0.892						
ESR	1.022	1.002, 1.042	0.029	1.012	0.984, 1.040	0.415	1.016	0.984, 1.050	0.324
CRP	1.019	1.001, 1.036	0.034	1.006	0.976, 1.037	0.701	1.008	0.971, 1.046	0.680
cMDA	1.006	1.001, 1.011	0.023	1.010	1.000, 1.019	0.055			
cMDA ≥ 221.7 ng/mL	11.098	2.027, 60.745					24.076	2.422, 239.368	0.007

AAV: ANCA-associated vasculitis; ANCA: antineutrophil cytoplasmic antibody; cMDA: circulating malondialdehyde; HR: hazard ratio; CI: confidence interval; MPO: myeloperoxidase; P: perinuclear; PR3: proteinase 3; C: cytoplasmic; BVAS: the Birmingham vasculitis activity score; FFS: the five-factor score; ESR: erythrocyte sedimentation rate; CRP: C-reactive protein.

**Table 4 medicina-60-01182-t004:** Multivariable Cox analyses of variables with *p* < 0.05 in univariable analysis.

Variables	Multivariable (*with cMDA*)			Multivariable (*with cMDA ≥ 221.7* ng/mL)		
	HR	95% CI	*p* Value	HR	95% CI	*p* Value
ESR (mm/h)	1.015	0.990, 1.041	0.241	1.017	0.987, 1.048	0.268
CRP (mg/L)	1.007	0.984, 1.031	0.553	1.011	0.981, 1.042	0.482
cMDA (ng/mL)	1.005	1.000, 1.011	0.064			
cMDA ≥ 221.7 ng/mL				13.462	2.235, 81.067	0.005

cMDA: circulating malondialdehyde; HR: hazard ratio; CI: confidence interval; ESR: erythrocyte sedimentation rate; CRP: C-reactive protein.

## Data Availability

The dataset collected and/or analysed in the present study are avail-able on request from the corresponding author.

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
