# Peer review of "Circulating Malondialdehyde Is a Potential Biomarker for Predicting All-Cause Mortality during Follow-Up by Reflecting Comprehensive Inflammation at Diagnosis in Patients with Antineutrophil Cytoplasmic Antibody-Associated Vasculitis"

_medicina, 2024, doi:10.3390/medicina60071182_

Round 1

Reviewer 1 Report

Comments and Suggestions for Authors

The study analyses for the first time the potential of circulating MDA as a predictor of all cause mortality and as indicator reflecting inflammation at the diagnosis in patients with AAV. Consider emphasizing that this is a pilot study. 

Overall, the study is well done, with clearly presented methodolology and data analysis. However, there are some aspects that could be improved. 

Introduction: sufficient and clear. Consider using abbrevation for malondialdehyde throughout the text (e.g. cMDA)

Materials and methods:  The authors should include all inclusion and exclusion criteria instead of citing their previous work on patients with vasculitis (ref 13 and 14). Did the authors consider including other comorbidities (in addition to DM and hypertension) associated with all cause mortality?

Results: Please include duration of the follow-up of the whole group.

Table 1: Please specify all values in the table.

Table 2: Please explaine the rationale for correlating MDA with glucose, cholesterol, BUN, creatinine, serum protein and albumine. Otherwise, consider removing these parameters from the Table.

Consider including supp table 1 in the main text, not as supplementary file

Disscussion: 

First paragraph: BVAS and FFS are numerical scores which are precisely calculated, and not estimated. Please revise the last sentence.

Line234: this is only speculation. Also, there is a very weak correlation between cMDA and ESR (r=0.251) to conclude that cMDA reflects the magnitute of comprehensive inflammation. Also, there is no correlation between cMDA and CRP. 

Lines 241, 242, 243: Please include relevant reference for this statement. 

Lines 258, 259, 260: The statement is a bit unclear. Please provide more clear explanation for the figure S1. 

Also, please comment on the potential differences between different types of AAV. This sholud be also included as a potential limitatation of the study, especially considering heterogenity of the AAV. 

Author Response

Reviewer (1)’s comments

Manuscript number: medicina-3082256

Title: Circulating malondialdehyde is a potential biomarker for predicting all-cause mortality during follow-up by reflecting comprehensive inflammation at diagnosis in patients with antineutrophil cytoplasmic antibody-associated vasculitis

We appreciate your excellent review of our manuscript. Your valuable comments helped us to make a better revision.

The study analyses for the first time the potential of circulating MDA as a predictor of all-cause mortality and as indicator reflecting inflammation at the diagnosis in patients with AAV. Consider emphasizing that this is a pilot study.

Overall, the study is well done, with clearly presented methodology and data analysis. However, there are some aspects that could be improved.

(1) Introduction: sufficient and clear. Consider using abbreviation for malondialdehyde throughout the text (e.g. cMDA)

             According to your recommendation, we changed “circulating malondialdehyde” to “cMDA” in the whole manuscript, Tables 1, 2, and 3, and Supplementary Tables but not in Supplementary Figure 1.

             For instance, in the ABSTRACT section,

“Abstract: Background and Objectives: To investigate whether circulating malondialdehyde (cMDA) at diagnosis could contribute to estimating cross-sectional comprehensive inflammation or vasculitis activity and further predicting all-cause mortality during follow-up in patients with antineutrophil cytoplasmic antibody-associated vasculitis (AAV).” (Lines 18-21)

(2) Materials and methods: The authors should include all inclusion and exclusion criteria instead of citing their previous work on patients with vasculitis (ref 13 and 14). Did the authors consider including other comorbidities (in addition to DM and hypertension) associated with all-cause mortality?

             As you indicated, we added detailed inclusion criteria and amended the text in the METHODS section as below:

“We randomly selected 80 patients with AAV from the Severance Hospital AN-CA-associated VasculitidEs (SHAVE) cohort, an observational cohort of Korean patients with AAV according to the inclusion criteria as follows: first of all, patients were diagnosed with AAV at the Division of Rheumatology, Department of Internal Medicine, Yonsei University College of Medicine, and Severance Hospital for the first time. They met the 2007 European Medicines Agency algorithms for AAV, the revised 2012 International Chapel Hill Consensus Conference Nomenclature of Vasculitides, and the 2022 American College of Rheumatology/European Alliance of Associations for Rheumatology (ACR/EULAR) classification criteria for MPA, GPA, and GPA [4, 5, 9-12]. Their medical records were well-documented enough to collect detailed clinical, laboratory, radiological, and histological data not only at diagnosis but also during follow-up to the last visit. Particularly, laboratory and radiological test results were recognised only when their tests were performed within less than 2 weeks at the time of AAV diagnosis. They were followed up for at least six months after AAV diagnosis. They had no serious medical conditions mimicking AAV such as malignancies or severe infectious diseases requiring hospitalisation as described in the 2022 ACR/EULAR criteria at the time of AAV diagnosis [9-11]. They were not exposed to immunosuppressive drugs for AAV treatment within at least four weeks (one month) before AAV diagnosis; however, glucocorticoids, less than 10mg dose equivalent to prednisolone, were allowed for relieving AAV symptoms. Of the 80 patients screened, two were excluded due to errors in measuring cMDA; as such, data from 78 patients were ultimately included and analysed.” (Line 68-88)

             Also, we added the context regarding the IRB approval and consent forms in the METHODS section as below:

“The present study was approved by the Institutional Review Board (IRB) of Severance Hospital, Seoul, Republic of Korea (IRB number 4-2016-0901), and conducted in accordance with the Declaration of Helsinki. All patients in this study provided written informed consent upon enrolment in the SHAVE cohort (at the time of AAV diagnosis and blood sampling). The IRB waived the requirements for additional written informed consent when it was obtained upon enrolment into the SHAVE cohort.” (Lines 90-95)

Additionally, in terms of mortality-related comorbidities, we agree that those comorbidities should have been included in this study. Nevertheless, in this study, we could not include mortality-related comorbidities other than T2DM and hypertension at the time of AAV diagnosis or before it owing to the limitation of the study design of a retrospective study. We added this context to the text in the LIMITATION section as below:

“Owing to this limitation of the study design of a retrospective study, mortality-related comorbidities other than T2DM and hypertension at the time of AAV diagnosis or before it could not be included in this study.” (Lines 299-302)

(3) Results: Please include duration of the follow-up of the whole group.

The follow-up duration based on all-cause mortality was defined as the period from AAV diagnosis to death for deceased patients and the period from AAV diagnosis to the last visit for surviving patients, respectively. For deceased patients, death means the same as the last visit; therefore, in this study, the follow-up duration in all 78 patients was the same as the follow-up duration based on all-cause mortality.

To avoid the confusion, we amended the text in the METHODS section as below:

“Regarding data during follow-up, the follow-up duration based on all-cause mortality was defined as the period from AAV diagnosis to death for deceased patients and the period from AAV diagnosis to the last visit for surviving patients, respectively. For deceased patients, death means the same as the last visit; therefore, in this study, the follow-up duration in all 78 patients was the same as the follow-up duration based on all-cause mortality.” (Lines 106-111)

(4) Table 1: Please specify all values in the table.

             As you commented, we added the description regarding the values of variables to the text in the RESULTS section, which were listed in Table 1 but not mentioned in the RESULTS section as below:

“There were three ex-smokers (3.8%), and the median BMI was 22.4 kg/m2.” (Lines 140-141)

“For the laboratory results, the median values were as follows: white blood cell count, 7,710. 0/mm3; haemoglobin, 12.5 g/dL; platelet count, 241,000/mm3; fasting glucose, 94.5 mg/dL; total cholesterol, 175.5 mg/dL; blood urea nitrogen, 19.3 mg/dL; serum creatinine, 0.8 mg/dL; total serum protein, 6.8 g/dL; and serum albumin, 4.2 g/dL.” (Lines 145-149)

(5) Table 2: Please explain the rationale for correlating MDA with glucose, cholesterol, BUN, creatinine, serum protein and albumine. Otherwise, consider removing these parameters from the Table.

             We agree to your recommendation and removed the variables in laboratory results in Table 2 and amended the title of Table 2. In addition, we amended the text in the RESULTS section as below:

“3.2. Correlation analysis 

At diagnosis, cMDA was significantly correlated with only ESR (r = 0.251, P = 0.027). cMDA also tended to be correlated with CRP (r = 0.222) but there was no statistical significance. Conversely, no significant correlations of cMDA with age, MPO-ANCA and PR3-ANCA titres, BVAS, and FFS were observed (Table 2). Additionally, there were no significant differences in cMDA according to the presence of MPO-ANCA (or P-ANCA) (P = 0.579), or PR3-ANCA (or C-ANCA) (P = 0.531).”

Table 2. Correlation analysis of cMDA with continuous variables of demographic data, AAV-specific indices, and acute-phase reactants at diagnosis in patients with AAV.

Variables

cMDA

Correlation

Coefficient

(r)

P value

Demographic data

Age

0.049

0.668

Body mass index

−0.187

0.101

AAV-specific indices

MPO-ANCA titre

0.078

0.498

PR3-ANCA titre

0.006

0.959

BVAS

0.117

0.310

FFS

0.163

0.154

Acute-phase reactants

ESR

0.251

0.027

CRP

0.222

0.058

cMDA: circulating malondialdehyde; AAV: ANCA-associated vasculitis; ANCA: antineu-trophil cytoplasmic antibody; MPO: myeloperoxidase; P: perinuclear; PR3: proteinase 3; C: cyto-plasmic; BVAS: the Birmingham vasculitis activity score; FFS: the five-factor score; ESR: eryth-rocyte sedimentation rate; CRP: C-reactive protein (Lines 162-176)

(6) Consider including supp table 1 in the main text, not as supplementary file

             As your recommended, we changed “supplementary Table 1 (Table S1)” to “Table 4”, and added it to the manuscript

“(Table 4). Therefore, we tentatively conclude that cMDA may be better or at least compa-rable to ESR, CRP, and BVAS in predicting all-cause mortality during follow-up in pa-tients with AAV.”

Table 4. Multivariable Cox analyses of variables with P <0.05 in univariable analysis.

Variables

Multivariable

(with cMDA)

Multivariable

(with cMDA ≥221.7 ng/mL)

HR

95% CI

P value

HR

95% CI

P value

ESR (mm/hr)

1.015

0.990, 1.041

0.241

1.017

0.987, 1.048

0.268

CRP (mg/L)

1.007

0.984, 1.031

0.553

1.011

0.981, 1.042

0.482

cMDA (ng/mL)

1.005

1.000, 1.011

0.064

cMDA ≥221.7 ng/mL

13.462

2.235, 81.067

0.005

cMDA: circulating malondialdehyde; HR: hazard ratio; CI: confidence interval; ESR: erythrocyte sedimentation rate; CRP: C-reactive protein. (Lines 255-262)

(7) Discussion: First paragraph: BVAS and FFS are numerical scores which are precisely calculated, and not estimated. Please revise the last sentence.

             According to your indication, to avoid confusion that BVAS and FFS were recalculated in this study, we changed the word “estimating” to “reflecting” and amended the text in the DISCUSSION section as below: Additionally, we changed “estimating” to “reflecting” in the ABSTRACT, INTRODUCTION and DISCUSSION sections.

“Based on these results, it was concluded that circulating malondialdehyde at diagnosis may have the potential to act as a biomarker for predicting all-cause mortality during follow-up but not as that for reflecting cross-sectional BVAS or FFS collected at diagnosis in patients with AAV.” (Lines 226-229)

(8) Line234: this is only speculation. Also, there is a very weak correlation between cMDA and ESR (r=0.251) to conclude that cMDA reflects the magnitute of comprehensive inflammation. Also, there is no correlation between cMDA and CRP.

We agree with your comment that this is only a speculation but not a definitive conclusion. Therefore, we amended this text in the DISCUSSION section as below:

“Therefore, despite a weak correlation between cMDA and ESR, and only a tendency of correlation between cMDA and CRP, it could be reasonably speculated that cMDA at diagnosis could act as a biomarker for predicting all-cause mortality by reflecting the extent of comprehensive (non-specific) effects of the inflammation-related risk factors of all-cause mortality.” (Lines 241-245)

(9) Lines 241, 242, 243: Please include relevant reference for this statement.

             According to your recommendation, we added two relevant references as reference number 21 and 22 in the DISCUSSION and REFERENCES section as below:

“Of note, among the six variables, cMDA exhibited a superior association with all-cause mortality compared with ESR and CRP, which are variables that reflect the degree of cross-sectional comprehensive inflammation and are the theoretical background for predicting mortality [21, 22].” (Lines 247-251)

“21. Watson, J.; Whiting, P.; Salisbury, C.; Banks, J.; Hamilton, W. Raised inflammatory markers as a predictor of one-year mortality: a cohort study in primary care in the UK using electronic health record data. BMJ Open. 2020, 10, e036027.

  1. Fest, J.; Ruiter, R.; Mooijaart, S.P.; Ikram, M.A.; van Eijck, C.H.J.; Stricker, B.H. Erythrocyte sedimentation rate as an in-dependent prognostic marker for mortality: a prospective population-based cohort study. J Intern Med. 2019, 285, 341-348.” (Lines 393-396)

(10) Lines 258, 259, 260: The statement is a bit unclear. Please provide more clear explanation for the figure S1.

             As you pointed out, we revised the text in the DISCUSSION section as below:

“Whereas, in terms of ESR, ESR may reflect the degree of comprehensive inflammation in patients with AAV as usual; however, it cannot participate in or amplify the vicious circle consisting of cMDA, oxidative stress, and inflammatory signals at all. (Figure S1). This inference may explain the discrepancy between cMDA and ESR in their potential to pre-dict all-cause mortality in patients with AAV.” (Line 268-273)

Additionally, we amended Figure S1 as below:

(11) Also, please comment on the potential differences between different types of AAV. This should be also included as a potential limitation of the study, especially considering heterogeneity of the AAV.

As you mentioned, there was a need for subgroup analysis among MPA, GPA, and EGPA patients; however, due to the nature of the pilot study and the small number of patients, subgroup analysis could not be performed.

We added this context in the LIMITATION section as below:

“Although there was a need for subgroup analysis among MPA, GPA, and EGPA patients, due to the small number of patients in this pilot study, subgroup analysis could not be performed.” (Lines 304-306)

Reviewer 2 Report

Comments and Suggestions for Authors

Dear Authors!
thank you for the opportunity to review your manuscript. AAV are the severe rheumatic disease with poor outcome. The finding of outcomes measures and surrogate biomarkers is useful for the clinical practice.

In the manuscript Authors assessed the role of MA as a biomarkers of AAV activity and an outcome measure for deatths.

I have several queries 

1) Please provide what time the sera for MA was assessed: at disease onset or during disease course. It is unclear, please clarify. If the time point was different please provide the time since onset to sample collection

2) I think the univariate analysis will be useful in two separate groups: died and survived and MA>222 and less than 222. It is interesting the clinical and laboratorial profiles in these groups. Also need the data about the disease course because the MA was measured at inclusion and deaths occurred later. Did Authors possibility to measure MA close to deaths?

3) Authosr can calculate OR some significant predictors

4) In the abstract it is unclear the statement "Circulating malondialdehyde ≥221.7 ng/mL at diagnosis exhibited a higher relative risk (RR 12.4, P = 0.008) for all-cause mortality during follow-up and a reduced survival rate (P <0.001) than circulating malondialdehyde <221.7 ng/mL." If MA>221 the probability of deaths higher, it is clear, but why if MA<221 the survival rate also lower? May be higher? Please clarify and add explanation in the text

Author Response

Reviewer (2)’s comments

Manuscript number: medicina-3082256

Title: Circulating malondialdehyde is a potential biomarker for predicting all-cause mortality during follow-up by reflecting comprehensive inflammation at diagnosis in patients with antineutrophil cytoplasmic antibody-associated vasculitis

We appreciate your excellent review of our manuscript. Your valuable comments helped us to make a better revision.

Dear Authors!

thank you for the opportunity to review your manuscript. AAV are the severe rheumatic disease with poor outcome. The finding of outcomes measures and surrogate biomarkers is useful for the clinical practice.

In the manuscript Authors assessed the role of MA as a biomarker of AAV activity and an outcome measure for deaths.

I have several queries

(1) Please provide what time the sera for MA was assessed: at disease onset or during disease course. It is unclear, please clarify. If the time point was different please provide the time since onset to sample collection

             Blood samples were collected from all patients at the time of AAV diagnosis. On the same day, they also gave consent for the enrollment in the cohort of AAV and for providing their blood.

             To clarify the time when blood samples were collected we amended the text in the METHODS section as below:

“Whole blood samples were collected from all patients at the time of diagnosis of AAV when written informed consent was obtained.” (Lines 117-118)

(2) I think the univariate analysis will be useful in two separate groups: died and survived and MA>222 and less than 222. It is interesting the clinical and laboratorial profiles in these groups.

             As you recommended, we added New Supplementary Tables 1 and 2 and their contexts in the text in the DISCUSSION section below:

“On the other hand, to validate our inferences, first, we compared the variables considered to be related to all-cause mortality between surviving and deceased patients. Deceased patients were older than surviving ones, as was naturally expected (74.5 vs. 62.5 years, P = 0.028). ESR (99.0 mm/h vs. 23.5 mm/h, P = 0.029), and CRP (51.6 mg/L vs. 3.0 mg/L, P = 0.035) levels were significantly elevated in deceased patients compared to surviving patients. Additionally, cMDA was also remarkably higher in deceased patients than in surviving patients (242.1 ng/mL vs. 92.9 ng/mL, P = 0.032) (Table S1). Second, when we compared the variables between patients with cMDA ≥221.7 ng/mL and those with cMDA <221.7 ng/mL, all-cause mortality was found more frequently in patients with cMDA ≥221.7 ng/mL than those without (28.6% vs. 3.1%, P = 0.008); however, no statistically significant differences in ESR and CRP levels were observed between the two groups (Table S2).To put it simply, these results showed that in the death-survival comparative analysis, both ESR and cMDA significantly contributed to all-cause mortality; however, in the cMDA cut-off-based comparative analysis, only death showed a significant difference, which is thought to support the inference that ESR might not participate in the cMDA vicious circle with a potential to make a significant contribution to an increase in the proportion of all-cause mortality. Although there was a significant difference in CRP levels between surviving and deceased patients, CRP and cMDA did not show a significant correlation, so they are not mentioned in the inference-supporting description section here.” (Line 274-293)

             In addition, we amended the text in “Supplementary Materials” section as below:

“Supplementary Materials: The following supporting information can be downloaded at: www.mdpi.com/xxx/s1, Figure S1: Inferences on the hypothesis of the clinical utility of cMDA in patients with AAV; Table S1: Comparison of variables between surviving and deceased pa-tients with AAV; Table S2: Comparison of variables between patients with cMDA ≥221.7 ng/mL and those with cMDA <221.7 ng/mL at diagnosis.” (Lines 318-322)

Also need the data about the disease course because the MA was measured at inclusion and deaths occurred later. Did Authors possibility to measure MA close to deaths?

             We agree with your indication; however, serial results of cMDA during the disease course, particularly, at the time close to death, were not available in this study. We added this context to the LIMITATION section as below:

“Last, serial results of cMDA during the disease course, particularly, at the time close to death, were not available in this study. However, as a pilot study, this investigation may have clinical significance because it explored possibilities for future studies. We believe that a future prospective study including more patients and concretely assessing inflammatory signals will provide more reliable and dynamic information regarding the clinical implications of cMDA at diagnosis in patients newly diagnosed with AAV.” (Lines 307-313)

(3) Authors can calculate OR some significant predictors

             In this study, univariable and/or multivariable logistic regression analyses of cMDA for all-cause mortality were not performed, and thus, we did not calculate ORs of predictors for mortality in this study. Instead, univariable, and multivariable Cox proportional hazards model analyses, and thus, we obtained HRs of independent predictors for mortality in Tables 3 and 4.

(4) In the abstract it is unclear the statement "Circulating malondialdehyde ≥221.7 ng/mL at diagnosis exhibited a higher relative risk (RR 12.4, P = 0.008) for all-cause mortality during follow-up and a reduced survival rate (P <0.001) than circulating malondialdehyde <221.7 ng/mL." If MA>221 the probability of deaths higher, it is clear, but why if MA<221 the survival rate also lower? May be higher? Please clarify and add explanation in the text

             As you indicated, we amended the texts in the ABSTRACT section as below:

“Compared to patients with cMDA <221.7 ng/mL, those with cMDA ≥221.7 ng/mL at diagnosis exhibited an increased relative risk (RR 12.4) for all-cause mortality, and further showed a decreased cumulative patient’s survival rate.” (Lines 28-30)

Round 2

Reviewer 2 Report

Comments and Suggestions for Authors

Dear Authors!

Thank you for the revised version of the manuscript

It has become better

I have no additional comments

Manuscri[t might be accepted ion the current form